# A Combined Biomarker That Includes Plasma Fibroblast Growth Factor 23, Erythropoietin, and Klotho Predicts Short- and Long-Term Morbimortality and Development of Chronic Kidney Disease in Critical Care Patients with Sepsis: A Prospective Cohort

**DOI:** 10.3390/biom13101481

**Published:** 2023-10-03

**Authors:** Luis Toro, Verónica Rojas, Carolina Conejeros, Patricia Ayala, Alfredo Parra-Lucares, Francisca Ahumada, Paula Almeida, María Fernanda Silva, Karin Bravo, Catalina Pumarino, Ana María Tong, María Eugenia Pinto, Carlos Romero, Luis Michea

**Affiliations:** 1Division of Nephrology, Department of Medicine, Hospital Clínico Universidad de Chile, Santiago 8380456, Chile; ltoro@med.uchile.cl (L.T.);; 2Centro de Investigación Clínica Avanzada, Hospital Clínico Universidad de Chile, Santiago 8380456, Chile; 3Centro de Pacientes Críticos, Clinica Las Condes, Santiago 7591047, Chile; 4Unidad de Pacientes Críticos, Departamento de Medicina, Hospital Clínico Universidad de Chile, Santiago 8380456, Chile; 5Instituto de Ciencias Biomédicas, Facultad de Medicina, Universidad de Chile, Santiago 8380456, Chile; 6Clinical Laboratory, Hospital Clínico Universidad de Chile, Santiago 8380456, Chile; 7Laboratorio de Fisiologia Integrativa, Facultad de Medicina Universidad de Chile, Santiago 8380456, Chile

**Keywords:** acute kidney injury, chronic renal insufficiency, sepsis, mortality, cohort studies

## Abstract

Acute Kidney Injury (AKI) is a frequent complication in intensive care unit (ICU) patients that increases mortality and chronic kidney disease (CKD) development. AKI is associated with elevated plasma fibroblast growth factor 23 (FGF23), which can be modulated by erythropoietin (EPO) and Klotho. We aimed to evaluate whether a combined biomarker that includes these molecules predicted short-/long-term outcomes. We performed a prospective cohort of ICU patients with sepsis and previously normal renal function. They were followed during their inpatient stay and for one year after admission. We measured plasma FGF23, EPO, and Klotho levels at admission and calculated a combined biomarker (FEK). A total of 164 patients were recruited. Of these, 50 (30.5%) had AKI at admission, and 55 (33.5%) developed AKI within 48 h. Patients with AKI at admission and those who developed AKI within 48 h had 12- and 5-fold higher FEK values than non-AKI patients, respectively. Additionally, patients with higher FEK values had increased 1-year mortality (41.9% vs. 18.6%, *p* = 0.003) and CKD progression (26.2% vs. 8.3%, *p* = 0.023). Our data suggest that the FEK indicator predicts the risk of AKI, short-/long-term mortality, and CKD progression in ICU patients with sepsis. This new indicator can improve clinical outcome prediction and guide early therapeutic strategies.

## 1. Introduction

Acute kidney injury (AKI) is a syndrome caused by a rapid decline in renal function [1]. AKI incidence in hospitalized adults is approximately 20%, rising to over 40% among intensive care unit (ICU) patients [1,2,3,4]. AKI is an independent risk factor for in-hospital mortality, and is associated with adverse long-term outcomes, including up to a five-fold increase in the risk of death at 1 year [1,5,6] and progression to chronic kidney disease (CKD), which is present in 25% of surviving AKI patients one year after admission to the ICU [1,7,8]. Sepsis, a life-threatening organ dysfunction caused by a dysregulated host response to infection [9], is the most common individual cause of AKI. Sepsis is associated with up to 50% of AKI cases, with up to 60% of sepsis patients having AKI, and the development of sepsis associated with AKI reduces survival and worsens short-term and long-term outcomes [10,11].

AKI is diagnosed by detecting elevated serum creatinine and/or reduced urine output [12]. However, these functional criteria have several limitations, including a delay between the onset of renal dysfunction and creatinine accumulation and low sensitivity and specificity for AKI [13]. New biomarkers for AKI that permit early detection of renal stress may overcome these limitations and serve as tools for predicting incident episodes or progression of AKI in susceptible patients exposed to harmful events. Indeed, recent studies in surgical and critically ill patients have explored the performance of available biomarkers to guide prevention and management [4,14,15,16,17,18].

In the present study, we evaluated the capacity to predict AKI, mortality, as well as other short-/long-term outcomes of a combined biomarker that included three plasma proteins: fibroblast growth factor-23 (FGF23), Klotho and erythropoietin (EPO) [19,20,21]. FGF23 is a hormone secreted primarily in bone tissue that promotes renal phosphate excretion [22,23], and Klotho acts as a cofactor for activating FGF23 receptors that are expressed and secreted in distal and proximal tubules [19,24]. Acute kidney injury (AKI) is a state of high FGF23 and low Klotho [19,20,21]. Recent studies have shown an early increase in blood levels of FGF23 in ICU patients who had developed AKI [25,26,27]. Klotho renal abundance was lower in the biopsy specimens of AKI patients [28,29], and the urine levels of Klotho decreased in hospitalized patients [30] and critically ill patients that had developed AKI [27]. EPO is mainly synthesized in the kidney under the transcriptional control of the hypoxia-inducible factor pathway, which responds to hypoxia and anemia [31]. Interestingly, EPO is a newly discovered inducer of bone FGF23 expression/secretion [32]. Plasma levels of EPO increase in critically ill patients that have developed AKI [26,33], and the rapid increase in circulating EPO caused by experimental AKI contributes to the increase in FGF23 [26,34].

We hypothesized that a combined indicator that includes plasma FGF23, EPO, and Klotho would predict the development of AKI, mortality, and long-term CKD progression in sepsis patients. Therefore, we studied a prospective cohort of septic ICU patients with previously normal renal function, measuring plasma FGF23, EPO, and Klotho levels at ICU admission to calculate the combined biomarker. We followed those patients throughout their hospitalization and one year after admission to determine the association between the combined biomarker and clinical outcomes, including inpatient AKI, short- and long-term mortality, and CKD progression.

## 2. Materials and Methods

### 2.1. Study Design and Setting

We performed a prospective cohort of septic patients admitted to the Hospital Clínico Universidad de Chile intensive care units. We evaluated the capacity of the combined biomarker to predict short-/long-term clinical outcomes, including AKI, mortality, and CKD at one year. The hospital Ethics Committee approved this study. Written informed consent was obtained from all patients or their representatives if they were not able to deliver consent at admission.

### 2.2. Cohort Characteristics

The patients included in the study were adults (>18 years) derived from the Emergency Room and admitted to the ICU with a diagnosis of severe sepsis/septic shock (Sepsis-2 definition) [35]. The rationale for using these definitions was that the study was designed in 2015, before the publication of the Third International Consensus Definitions for Sepsis and Septic Shock (Sepsis-3 criteria) in 2016 [35], and patient recruitment started before the implementation of the current sepsis guidelines in our ICU.

All patients had previous outpatient serum creatinine measurements (up to 6 months before admission), which were used to calculate baseline estimated glomerular filtration rate (eGFR) with the CKD-EPI formula. If a patient had more than one measurement, we used the value obtained closest to their admission to the ICU. Only patients with eGFR > 60 mL/min/1.7 m^2^ were included in the study. Patients with prior CKD, a history of renal transplant, or current pregnancy were excluded.

### 2.3. Outcomes

The primary outcome was one-year mortality. Secondary short-term outcomes included the development of AKI within 24–48 h of admission (defined according to KDIGO criteria for change in serum creatinine vs. baseline), need for RRT, vasoactive drugs, or mechanical ventilation (MV) up to 30 days after admission. To identify AKI, all patients had at least three serum creatinine measurements (admission, 24 h, and 48 h); patients were classified as AKI at admission if they had serum creatinine levels 0.3 mg/dL or more higher compared to the baseline value, and to define AKI during the first 48 h, we considered this to be the case for patients with an increase of 0.3 mg/dL or above at 24 or 48 h (as compared to the baseline value) without increased values at admission. The highest serum creatinine value was used to classify AKI severity (using KDIGO criteria). Secondary long-term outcomes included mortality at 6 months, CKD progression (defined as eGFR < 60 mL/min/1.7m^2^, using CKD-EPI formula), and need for chronic RRT at one year. To define CKD, we used outpatient serum creatinine measurements obtained between 12 and 15 months after admission to the ICU; if the patient had two or more measurements, we considered the highest serum creatinine when estimating eGFR.

### 2.4. Data Collection and Plasma Measurements

Venous blood samples were obtained to measure serum creatinine, plasma FGF23, EPO, and Klotho during the first six hours after ICU admission. The samples were transported to a clinical laboratory and centrifuged at 3000 rpm for 5 min. Plasma was extracted and stored at −80 °C until measurements were performed. Demographic, clinical, and biochemical data were collected during the hospitalization. In addition, blood samples from surviving patients were obtained one year after admission to measure serum creatinine and eGFR (CKD-EPI formula). Biochemical parameters were measured with a Roche/Hitachi Modular-DP Analyzer (Roche Diagnostics, Basel, Switzerland). Plasma FGF23, EPO, and Klotho measurements were performed using ELISA (intact FGF23: 60-6600, Quidel Immutopics, San Diego, CA; EPO: DEP00, R&D Systems, Minneapolis, MN. Klotho: 27998, IBL-America, Minneapolis, MN, USA) according to manufacturer’s specifications. The inter-assay coefficient of variation for FGF23 was between 3.5 and 9.1%, for EPO between 4.2 and 8.2%, and for Klotho between 2.9 and 11.4%.

### 2.5. Design of Combined Biomarker

We designed a combined biomarker named FEK using plasma FGF23, EPO, and Klotho data. The formula of this biomarker is presented below.
(1)$$FEK=\frac{\left[FGF23\right]p*\left[EPO\right]p}{\left[Klotho\right]p}$$

The rationale for this formula was based on previous data [21,30] showing that FGF23 and EPO are elevated in AKI patients, while Klotho levels are reduced. We determined FEK values for all patients and evaluated its predictive capacity for the onset of AKI within 48 h of admission, mortality, and other outcomes.

### 2.6. Sample Size Calculation

The sample size calculated for the clinical study was 164 patients. This number allowed us to detect a minimum one-year mortality hazard ratio (HR) of 1.6 between patients with high vs. low values. This HR was determined using preliminary data from our group, in which we evaluated 45 ICU patients with or without AKI and observed a HR >1.6 for one-year mortality in the patients with higher values. We assumed 10% attrition, 5% alpha error, and 80% power.

### 2.7. Statistical Analysis

Fisher’s exact test was used for categorical variables. The Shapiro–Wilk test was applied to determine whether continuous variables were normally distributed. Parametric variables were expressed as arithmetic mean ± standard deviation, and non-parametric variables as median [percentile 25–75]. For parametric variables, we used Student’s *t*-test (2 groups) and one-way ANOVA with Tukey’s post hoc test (more than 2 groups). For non-parametric variables, we used the Mann–Whitney U (2 groups) and Kruskal–Wallis tests with Dunn’s post hoc test (more than 2 groups). To determine the diagnostic accuracy of the FEK biomarker for predicting the development of AKI within 48 h of admission, a receiver operating characteristic (ROC) analysis was performed, including calculation of the area under the curve (AUC) plus sensitivity/specificity of various cut-off values. To determine a diagnostic cut-off value, we used the value with the highest Youden’s Index (J) (sensitivity + specificity − 1). Survival analysis was performed using the Mantel–Haenszel method, including hazard ratio (HR) calculation, 95% confidence intervals (95% CI), and Kaplan–Meier curves. Multivariate analyses were performed using multivariate logistic regression. The results are expressed as HR with 95%CI using Forest Plot representations. Data were analyzed using GraphPad Prism v.6.0 (GraphPad Software, La Jolla, CA) and Stata/SE 15.0 software (Stata Software, College Station, TX, USA). All analyses were two-tailed, and *p* < 0.05 was defined as a statistically significant difference.

## 3. Results

### 3.1. Patient Characteristics

Between June 2016 and October 2018, 814 patients with severe sepsis/septic shock admitted to the ICUs from the Emergency Room were evaluated, and 164 patients were recruited into the study (Figure 1). The cohort was classified into three groups according to the presence of AKI: Group 1 (patients with AKI at admission), n = 50; Group 2 (patients who developed AKI within 48 h of admission), n = 55; and Group 3 (patients who did not develop AKI), n = 59. Baseline characteristics are presented in Table 1.

### 3.2. FEK Indicator and Prediction of AKI

The combined FEK indicator was higher in both AKI groups than in non-AKI patients (Group 1: 4.33 [2.46–9.61]; Group 2: 2.01 [1.21–3.03], *p* < 0.05 vs. Group 1; Group 3: 0.58 [0.25–0.78], *p* < 0.001 vs. Groups 1 and 2) (Figure 2a; Table 2). Moreover, the increases in FEK values were positively correlated with AKI severity (Figure 2b). The fold differences in the FEK indicator for AKI vs. non-AKI patients were significantly greater than the differences in plasma FGF23, EPO, or Klotho alone. 

We evaluated the diagnostic accuracy of FEK in predicting AKI during the first 48 h in patients without increased serum creatinine levels at admission (Group 2 vs. 3). We found that FEK was highly accurate, with an AUC: 0.87 [0.80–0.93] (Figure 2c). This AUC was higher that serum creatinine at admission (AUC: 0.58 [0.47–0.68]). We determined a cut-off point, defined as the value with the highest Youden Index [sensitivity + specificity − 1] for AKI prediction, which was an FEK value of 0.87 A.U.

### 3.3. FEK Indicator and Outcome Prediction

To evaluate the predictive capacity of the indicator for other acute outcomes, we compared patients with FEK values at admission below vs. equal to or above the cut-off point (Table 3). We found that patients with FEK values above the cut-off were more likely to require RRT (21.0% vs. 0.0%; *p* < 0.001) and vasoactive drugs (67.6% vs. 37.3%; *p* < 0.001) and had higher 30-day mortality (26.7% vs. 11.9%; *p* = 0.024) (Figure 3a).

Finally, we evaluated the capacity of FEK to predict long-term outcomes. We found that patients with FEK levels above the cut-off value had increased 1-year mortality (41.9% vs. 18.6%; *p* = 0.003; H.R.: 2.30) and higher rates of CKD (26.2% vs. 8.3%; *p* = 0.023; H.R.: 3.91; Figure 3b). Additionally, the percentage of surviving patients without CKD at one year was lower in patients with FEK values above the cut-off at admission than in patients below the cut-off value (42.9% vs. 74.6%, *p* < 0.001). Multivariate analysis indicated that increased FEK values, adjusted by other potential covariables, including SOFA score and baseline variables, are independent predictors of 30-day and 1-year mortality (Figure 4).

## 4. Discussion

We evaluated FGF23, EPO, and Klotho plasma levels as a combined biomarker for AKI prediction and prognosis. Septic patients admitted to ICU with AKI and patients who developed AKI within 24–48 h had higher FEK values at admission. In addition, FEK values above the calculated threshold predicted AKI development in patients with serum creatinine levels within the normal range at admission. Moreover, elevated FEK levels at admission were correlated with AKI severity and worse short-term outcomes, including mortality. Finally, FEK values above the cut-off were associated with worse long-term outcomes, such as 1-year mortality and CKD development. These results suggest that FEK is a promising tool for identifying critically ill patients at risk of developing an AKI episode and a biomarker of long-term risk for post-ICU survivors.

Plasma FGF23 comprises the bioactive intact hormone (iFGF23) and N-terminal and C-terminal fragments generated by cleavage, and both iFGF23 and C-terminal FGF23 fragment (cFGF23) can be modified by renal injury [20,23]. The finding of higher values of iFGF23 in patients with AKI admitted to the ICU or in patients that developed AKI up to 48 h post admission is consistent with previous studies that have shown higher levels of plasma FGF23 in samples from critically ill and cardiac surgery patients taken 24–48 h before the diagnosis of AKI [25,36,37,38]. High FGF23 levels are an essential determinant of FEK values above the cut-off, and the increase in EPO and the decrease in Klotho showed a significant correlation with changes in iFGF23. However, although every single biomarker showed a significant difference in patients that developed moderate and/or severe AKI compared to non-AKI patients, the combined biomarker showed changes of greater magnitude and could also identify patients that developed mild AKI. Further studies, including other types of AKI and/or patients with CKD before the injury, are necessary to evaluate the performance of the FEK vs. single biomarkers.

Knowledge of the mechanisms explaining plasma FGF23 increase in AKI is informative for the rational selection and potential use of biomarkers [26,39,40]. Decreased renal clearance due to renal dysfunction [41], in addition to higher synthesis in the liver, thymus, spleen, and bone due to non-classic regulators of FGF23 production (PTH, vitamin D, and phosphate), would explain the acute increase in plasma FGF23 in AKI [20,21]. In sepsis, FGF23 expression and secretion can be induced by cytokines [42] and by signals triggered by renal hypoxia, such as glycerol-3-phosphate and EPO [26,40]. EPO directly increases FGF23 production and cleavage in bone marrow via its homodimeric receptor (EPOR) [26,34,43,44,45]. In an experimental model of septic AKI, we found that plasma EPO increased very early after injury, induced bone marrow expression of FGF23 via EPOR, and that the blockade of EPOR partially prevented the increase in plasma levels of FGF23 caused by AKI [26,46]. In line with these results, in the present study, we found higher EPO levels at admission correlated with higher plasma iFGF23. Another study also showed that critically ill patients who developed AKI had higher EPO levels at ICU admission [47], and Rabadi et al. found a positive association between the number of red blood cell transfusions, an indirect indicator of acute blood loss, and plasma cFGF23 levels [43]. Previous studies of sepsis have reported altered renal blood flow (RBF) and an association between systemic hemodynamic instability and renal dysfunction in septic shock [48] with decreased renal oxygenation [49]. Thus, detecting an early rise in plasma EPO as part of the biomarker may be informative of renal hypoxia, which is a factor contributing to FGF23 secretion.

Our results also showed lower plasma Klotho levels in septic patients admitted with AKI or who developed AKI within 24–48 h. In line with these results, a cohort study of cardiac surgery patients showed an acute drop in Klotho plasma levels in patients that developed AKI [50]. An acute reduction in renal Klotho mRNA and protein abundance has been detected in several experimental models of AKI, including sepsis by cecal ligation and puncture (CLP) and lipopolysaccharide (LPS) injection [29,51,52], and the reduction in plasma Klotho paralleled the decrease in renal Klotho mRNA/protein in experimental AKI [30]. Renal biopsy specimens from AKI patients diagnosed with acute tubular necrosis or acute tubulointerstitial nephritis showed lower Klotho protein abundance to be correlated with AKI severity [28]. Moreover, the analysis of post-mortem sepsis–acute kidney injury biopsies showed a significant reduction in the abundance of Klotho mRNA [30], and a study in critically ill patients found lower levels of urine Klotho (normalized by urine creatinine) in samples obtained 24–48 h after AKI diagnosis [27]. Thus, an early reduction in circulating Klotho would be a common feature in patients that develop AKI, and the decrease in plasma Klotho would result from reduced renal Klotho expression. Studies in experimental AKI indicate that Klotho reduction is caused by reduced synthesis/secretion, secondary to tubular cell damage [30]. In addition, oxidative stress and proinflammatory cytokines such as TNF-alpha can reduce Klotho protein abundance [53,54,55]. Interestingly, Klotho deficiency aggravates experimental sepsis-related multiorgan function [51,56], and it has been proposed that Klotho reduction may be a pathogenic factor, promoting inflammation, senescence, and maladaptive recovery following AKI [51,57,58].

Although it is recognized that post-AKI patients have increased long-term mortality and complications, including a higher risk of CKD [6,8,11], very few studies are available identifying biomarkers predicting outcomes following an AKI episode [59]. A multi-center international study to identify biomarkers of the persistence of KDIGO stage 3 AKI in ICU patients with AKI KDIGO stage 2 or 3 found that higher levels of urinary C-C motif chemokine ligand 14 (CCL-14) predicted the persistence of renal dysfunction [60]. A combination of urinary biomarkers for tubular and cortical tubular damage, ganglioside M2 activator protein (GM2AP), and tail-less complex polypeptide-1 eta subunit (TCP1-eta), respectively, predicted recovery from AKI with a success rate of around 80% [61]. In the present study, we found that FEK levels above the threshold at admission were correlated with worse long-term outcomes, including 1-year mortality and CKD progression, suggesting that FEK may help identify patients at higher risk and for the guidance of preventive management of progression to CKD after AKI. Further studies will be necessary to characterize the evolution of FEK components and analyze their pattern of evolution and the association with long-term outcomes. Moreover, validated biomarkers predictive of long-term outcomes are needed.

Our clinical study has limitations. The original study was performed in the intensive care unit of a single hospital (i.e., it was a unicenter study). Furthermore, the study did not evaluate other plasma biomarkers. While FEK was highly accurate at diagnosing AKI, further studies are needed to determine whether FEK has a similar or better predictive capacity than other biomarkers. Another limitation is that additional variables were not included in our analysis, such as body mass index, serum bicarbonate, and additional severity parameters besides SOFA. In addition, this study included only patients with documented, previously normal eGFR values. It has been suggested that CKD patients admitted to the ICU should be treated similarly to AKI patients [12], independent of serum creatinine levels. Thus, patients with normal renal function before hospitalization would be the group with the highest potential to benefit from medical interventions guided by an AKI biomarker such as FEK. Nevertheless, future studies should be performed to evaluate whether the FEK biomarker can predict AKI and clinical outcomes in patients with prior history of CKD.

Another limitation is that our study included patients based on the Sepsis-2 criteria (severe sepsis and septic shock) [35]. The current criteria used in ICU patients are the Sepsis-3 criteria, which define sepsis as life-threatening organ dysfunction caused by a dysregulated host response to infection and considers the quick Sequential Organ Failure Assessment (qSOFA) scoring system [9]. Observational studies indicate differences in clinical characteristics and outcomes in patients using Sepsis-2 versus Sepsis-3 criteria [62,63]. Therefore, a new study is required to evaluate the accuracy of the FEK biomarker in septic patients using the current criteria.

## 5. Conclusions

The combined FEK indicator, including Fibroblast Growth Factor 23, erythropoietin, and Klotho, predicted the development of AKI, short-/long-term mortality, and one-year CKD progression in critical care patients with sepsis. To the best of our knowledge, this is the first study showing that a combined biomarker consisting of FGF23–EPO–Klotho has the potential to improve the prediction of clinical outcomes and guide therapeutic strategies in AKI patients. However, the indicator should be evaluated in a large multicenter clinical study that includes a comparison with other AKI biomarkers.

## Figures and Tables

**Figure 1 biomolecules-13-01481-f001:**
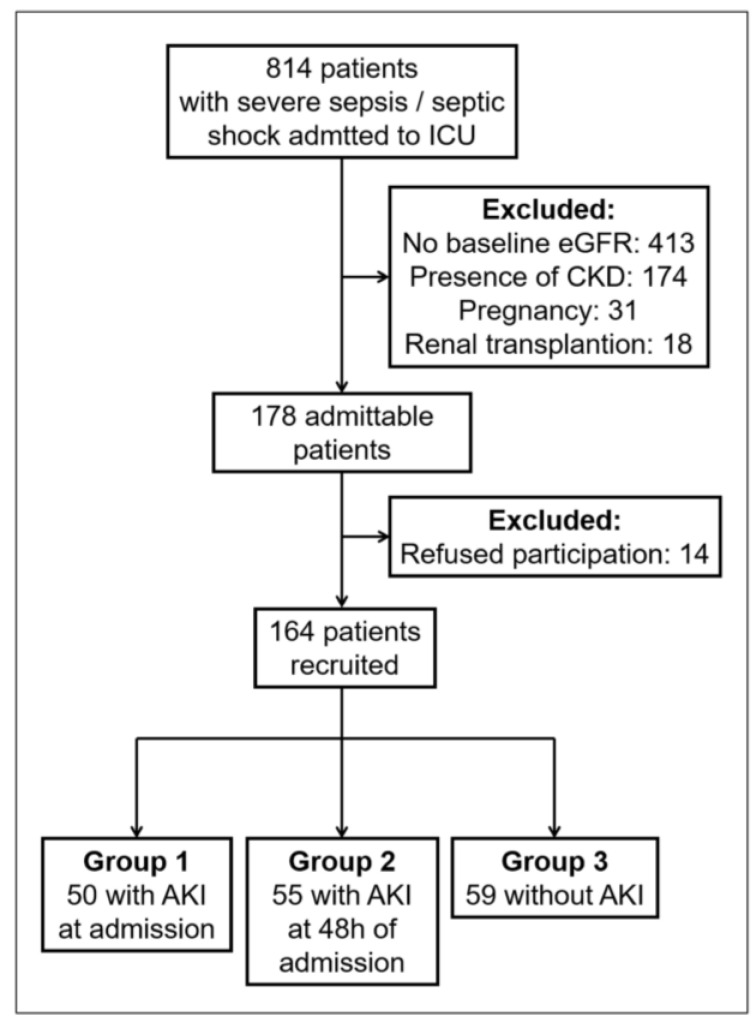
Patient selection flowchart. ICU = intensive care unit; eGFR = estimated glomerular filtration rate; CKD = chronic kidney disease; AKI = acute kidney injury.

**Figure 2 biomolecules-13-01481-f002:**
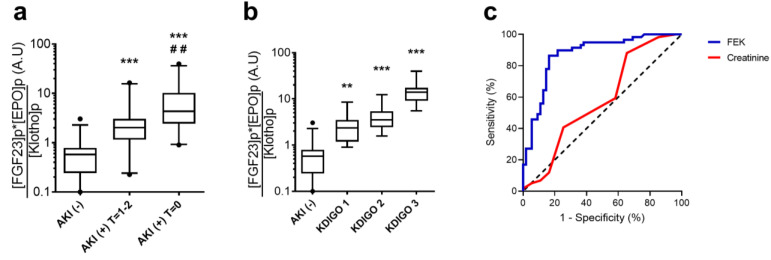
Combined FEK indicator at admission predicts AKI in septic ICU patients. (**a**,**b**) FEK values of patients at admission, stratified by the presence and time to onset of AKI. AKI (+) T = 0: Patients with AKI at admission; AKI (+) T = 1–2: patients who developed AKI within 48 h of admission; AKI (−): patients who did not develop AKI. (**a**) Boxplot of the combined FEK indicator at ICU admission (data presented in logarithmic scale). (**b**) Boxplot of the combined FEK indicator of AKI (+) T = 0 patients, stratified by the severity of AKI defined by the KDIGO criteria (data presented in logarithmic scale). (**c**) Receiver Operating Characteristic (ROC) analysis of the combined FEK indicator at admission (blue line) versus serum creatinine at admission (red line) for predicting the development of AKI within 48 h of admission in septic ICU patients with serum creatinine within normal range at admission; FEK: 0.87 [0.80–0.93], serum creatinine: 0.58 [0.47–0.68]. AKI (+) T = 0: n = 50; AKI (+) T = 1–2: n = 55; AKI (−): n = 59. ** *p* < 0.01, *** *p* < 0.001 versus AKI (−); # # *p* < 0.05 vs. AKI (+) T = 1–2. Black points are outliers.

**Figure 3 biomolecules-13-01481-f003:**
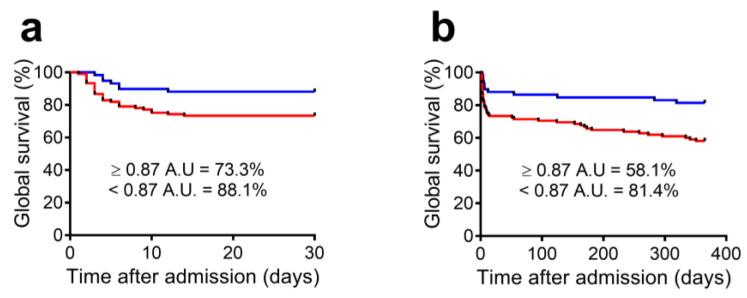
Admission FEK levels predict short- and long-term mortality in septic ICU patients, stratified by high or low FEK value (cut-off = 0.87 A.U.). Kaplan–Meier curves for (**a**) 30-day and (**b**) 1-year global survival. Absolute percentages are shown (red line: survival of patients with FEK levels equal or above cut-off; blue line: survival of patients with FEK levels below cut-off).

**Figure 4 biomolecules-13-01481-f004:**
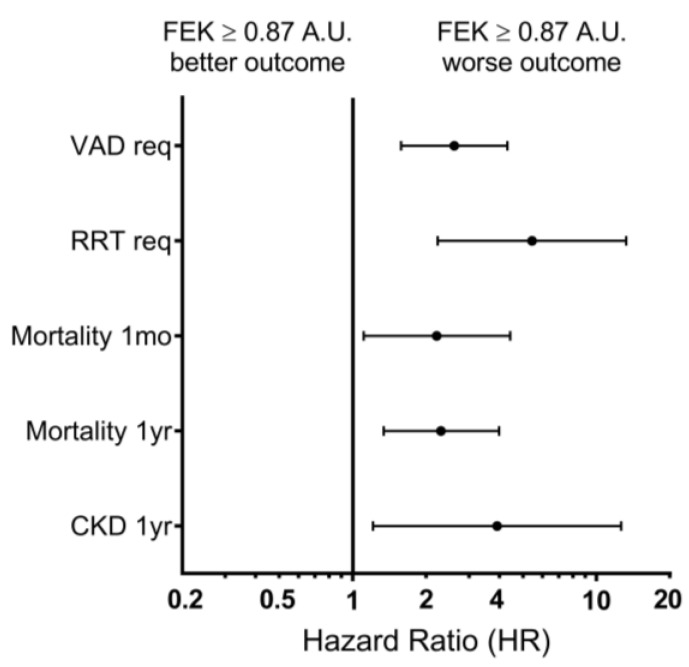
Forest plot graph with adjusted hazard ratios (HR) with the 95% confidence interval (data presented in logarithmic scale) of short- and long-term outcomes, stratified by high or low FEK value (cut-off = 0.87 A.U.). VAD req = vasoactive drugs requirements; MV req = mechanical ventilation requirements; RRT req = renal replacement therapy requirements; CKD 1 yr = development of chronic kidney disease at 1 year after admission.

**Table 1 biomolecules-13-01481-t001:** Demographic and clinical characteristics of patients enrolled in the study. eGFR = estimated glomerular filtration rate. ^#^ eGFR calculated by CKD-EPI formula. * *p* < 0.05; ** *p* < 0.01; *** *p* < 0.001 versus patients without AKI.

Characteristic		All Patients	Patients with AKI at Admission	Patients with AKI at Day 1–2	Patients without AKI
Demographic					
	Patients, n (%)	164 (100%)	50 (30.5%)	55 (33.5%)	59 (36.0%)
	Mean age ± SD, yr	63.4 ± 14.3	64.7 ± 16.6	63.7 ± 13.9	62.1 ± 12.6
	Men, n (%)	103 (62.8%)	33 (66.0%)	38 (69.1%)	32 (54.2%)
	Hispanic race, n (%)	164 (100%)	50 (100%)	55 (100%)	59 (100%)
	Hypertension, n (%)	92 (56.1%)	27 (54.0%)	32 (58.2%)	33 (55.9%)
	Diabetes, n (%)	35 (21.3%)	13 (26.0%)	11 (20.0%)	11 (18.6%)
	Coronary heart disease, n (%)	20 (12.2%)	7 (14.0%)	8 (14.5%)	5 (8.5%)
Baseline renal function					
	Serum creatinine ± SD, mg/dL	0.85 ± 0.16	0.86 ± 0.16	0.85 ± 0.18	0.84 ± 0.15
	eGFR ^#^ ± SD, mL/min/1.73m^2^	85.2 ± 15.5	83.9 ± 13.9	86.5 ± 15.8	85.1 ± 16.7
Infectious foci, n (%)					
	Pulmonary	60 (36.6%)	22 (44.0%)	20 (36.4%)	18 (30.5%)
	Urinary	26 (15.9%)	8 (16.0%)	10 (18.2%)	8 (13.6%)
	Abdominal	78 (47.6%)	20 (40.0%)	25 (45.5%)	33 (55.9%)
	Emergency surgery required	68 (41.5%)	19 (38.0%)	23 (41.8%)	26 (44.1%)
Biochemical parameters at admission					
	Hematocrit ± SD, g/dL	9.8 ± 2.1	9.3 ± 1.6	9.9 ± 1.8	10.9 ± 1.8
	Blood ureic nitrogen ± SD, mg/dL	24.6 ± 9.8	33.1 ± 10.4 ***	21.7 ± 7.1	20.1 ± 6.5
	Serum calcium ± SD, mg/dL	8.8 ± 1.1	8.5 ± 1.1 *	8.8 ± 0.9	9.0 ± 1.1
	Serum phosphate ± SD, mg/dL	3.9 ± 0.9	4.0 ± 0.9	4.0 ± 0.9	3.8 ± 0.9
	Serum magnesium ± SD, mg/dL	1.9 ± 0.2	1.8 ± 0.2	1.9 ± 0.2	2.0 ± 0.2
	Serum creatinine ± SD, mg/dL	1.35 ± 0.90	2.35 ± 1.07 ***	0.94 ± 0.21	0.89 ± 0.16
	Maximum serum creatinine ± SD, mg/dL	1.91 ± 1.08	2.69 ± 1.09 ***	2.22 ± 0.86 ***	0.95 ± 0.14
Clinical parameters at admission					
	Systolic arterial pressure ± SD, mmHg	104.5 ± 21.1	100.2 ± 20.5 *	103.9 ± 18.6	108.7 ± 23.2
	Diastolic arterial pressure ± SD, mmHg	55.9 ± 13.1	54.5 ± 13.3	54.8 ± 13.0	58.2 ± 12.8
	Mean arterial pressure ± SD, mmHg	72.1 ± 13.0	69.8 ± 13.5 *	71.1 ± 12.1	75.1 ± 13.0
	Total SOFA score	5 [3–7]	8 [6–10] ***	5 [4–7] **	4 [3–5]
	Non-renal SOFA score	5 [3–6]	6 [4–7] ***	5 [3–6] **	4 [3–5]
	Renal SOFA score	0 [0–1]	2 [1–3] ***	0 [0–0]	0 [0–0]
AKI severity (KDIGO criteria)—no. (%)					
	KDIGO 1	33 (20.1%)	15 (30.0%)	18 (32.7%)	0 (0.0%)
	KDIGO 2	38 (23.2%)	19 (38.0%)	19 (34.5%)	0 (0.0%)
	KDIGO 3	34 (20.7%)	16 (32.0%)	18 (32.7%)	0 (0.0%)

**Table 2 biomolecules-13-01481-t002:** Plasma FGF23, EPO, and Klotho values and combined FEK values for septic patients at admission. Data are expressed as median [percentile 25–75].

		FGF23 (pg/mL)	EPO (mIU/mL)	Klotho (mIU/mL)	F.E.K. (A.U.)
All patients (n = 164)		37.2 [24.0–56.8]	16.6 [12.3–21.5]	422.8 [284.4–504.6]	1.52 [0.71–3.47]
AKI (+) at admission (n = 50)		63.3 [48.5–83.0]	20.1 [15.9–25.5]	289.3 [194.4–368.7]	4.33 [2.46–9.61]
	KDIGO 1 (n = 15)	45.2 [36.8–52.5]	17.5 [14.1–20.4]	369.9 [312.8–437.6]	2.33 [1.29–3.16]
	KDIGO 2 (n = 19)	61.5 [49.9–73.3]	17.9 [14.9–20.6]	284.5 [236.7–353.6]	3.50 [2.54–5.22]
	KDIGO 3 (n = 16)	88.7 [75.5–96.6]	27.9 [24.1–32.0]	186.7 [152.2–199.2]	14.04 [9.61–17.26]
AKI (+) at day 1–2 (n = 55)		43.2 [34.6–54.1]	17.6 [14.6–23.8]	392.2 [275.3–505.2]	2.01 [1.21–3.03]
	KDIGO 1 (n = 18)	34.6 [32.0–37.1]	15.4 [12.1–19.9]	424.1 [375.4–563.7]	1.29 [1.05–1.57]
	KDIGO 2 (n = 19)	42.6 [36.5–49.6]	17.1 [14.8–22.4]	416.4 [301.1–487.7]	2.25 [1.51–2.90]
	KDIGO 3 (n = 18)	57.4 [44.7–79.3]	19.7 [15.6–34.9]	272.0 [197.4–490.2]	4.86 [2.44–11.54]
AKI (−) (n = 59)		21.3 [14.9–28.1]	12.3 [8.9–17.0]	493.1 [442.6–593.7]	0.58 [0.25–0.78]

**Table 3 biomolecules-13-01481-t003:** Demographic and clinical characteristics of patients enrolled in the study, stratified by the combined FEK indicator at admission (below or above the cut-off value – 0.87 A.U,). eGFR = estimated glomerular filtration rate. ^#^ eGFR calculated by CKD-EPI formula. * *p* < 0.05; *** *p* < 0.001 versus patients with FEK < 0.87 A.U. at admission.

Characteristic		All Patients	FEK < 0.87 A.U.	FEK ≥ 0.87 A.U.
Demographic				
	Patients, n (%)	164 (100%)	59 (36.0%)	105 (64.0%)
	Mean age ± SD, yr	63.4 ± 14.3	61.9 ± 11.4	64.3 ± 15.7
	Men, n (%)	103 (62.8%)	35 (59.3%)	68 (64.8%)
	Hypertension, n (%)	92 (56.1%)	34 (57.6%)	58 (55.2%)
	Diabetes, n (%)	35 (21.3%)	10 (16.9%)	25 (23.8%)
	Coronary heart disease, n (%)	20 (12.2%)	6 (10.2%)	14 (13.3%)
Baseline renal function				
	Serum creatinine ± SD, mg/dL	0.85 ± 0.16	0.84 ± 0.16	0.85 ± 0.17
	eGFR ^#^ ± SD, mL/min/1.73m^2^	85.2 ± 15.5	85.9 ± 16.4	84.9 ± 15.1
Infectious foci, n (%)				
	Pulmonary	60 (36.6%)	20 (33.9%)	40 (38.1%)
	Urinary	26 (15.9%)	8 (13.6%)	18 (17.1%)
	Abdominal	78 (47.6%)	31 (52.5%)	47 (44.8%)
	Emergency surgery required	68 (41.5%)	27 (45.8%)	41 (39.0%)
Biochemical parameters at admission				
	Blood ureic nitrogen ± SD, mg/dL	24.6 ± 9.8	20.6 ± 6.2	26.9 ± 10.7 ***
	Serum calcium ± SD, mg/dL	8.8 ± 1.1	8.9 ± 1.1	8.7 ± 1.0
	Serum phosphate ± SD, mg/dL	3.9 ± 0.9	3.8 ± 0.9	4.0 ± 0.9
	Serum creatinine ± SD, mg/dL	1.35 ± 0.90	0.90 ± 0.17	1.60 ± 1.04 ***
	Maximum serum creatinine ± SD, mg/dL	1.91 ± 1.08	1.07 ± 0.49	2.38 ± 1.03 ***
Clinical parameters at admission				
	Systolic arterial pressure ± SD, mmHg	104.5 ± 21.1	109.1 ± 23.0	101.9 ± 19.6 *
	Diastolic arterial pressure ± SD, mmHg	55.9 ± 13.1	58.7 ± 12.8	54.4 ± 13.0 *
	Mean arterial pressure ± SD, mmHg	72.1 ± 13.0	75.5 ± 13.0	70.2 ± 12.7 *
	Total SOFA score	5 [3–7]	4 [3–5]	6 [4–9] ***
	Non-renal SOFA score	5 [3–6]	4 [3–5]	5 [4–6] ***
	Renal SOFA score	0 [0–1]	0 [0–0]	1 [0–2] ***
AKI development—no. (%)				
	No AKI (−)	59 (36.0%)	53 (89.8%)	6 (5.7%) ***
	AKI at admission	50 (30.5%)	0 (0%)	50 (47.6%) ***
	AKI at day 1–2	55 (33.5%)	6 (10.2%)	49 (46.7%) ***

## Data Availability

The data that support our study are available upon request to the corresponding author. These data are not publicly available because they contain information that could potentially compromise participants’ privacy and local restrictions.

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
