# Peer review of "A Combined Biomarker That Includes Plasma Fibroblast Growth Factor 23, Erythropoietin, and Klotho Predicts Short- and Long-Term Morbimortality and Development of Chronic Kidney Disease in Critical Care Patients with Sepsis: A Prospective Cohort"

_biomolecules, 2023, doi:10.3390/biom13101481_

Round 1

Reviewer 1 Report

The study is very interesting, especially the development of the FEK value.

In the future, I would recommend using a state-of-the-art method, such as Luminex, for protein quantification.

For this job, I really have nothing to add. Congrats

Reviewer 2 Report

AKI is associated with elevated plasma fibroblast growth factor 23(FGF 23), which can be modulated by erythropoietin (EPO) and Klotho. Authors aimed to evaluate whether a combined biomarker that includes these molecules predicted short/long-term outcomes. Their data suggest that the FEK indicator predicts the risk of AKI, short/long-term mortality, and CKD progression in ICU patients with sepsis.

In order to employ in clinical practice in the future, it is necessary to be superior to existing biomarkers, so it would be better to compare it with existing data. For example, comparison with tubular markers such as NGAL and LFABP would be very interesting.

 In recent years, there is a concept of acute kidney disease(AKD), which describes the state of transition from AKI to CKD (Nat Rev Nephrol. 2017 Apr;13(4):241-257.). I think it would be important if the present biomarker could show a correlation with serum creatinine levels after three months in this study.

Reviewer 3 Report

I reviewed the manuscript by Toro et al titled ‘A combined Biomarker That Includes Plasma Fibroblast 2 Growth Factor 23, Erythropoietin, and Klotho Predicts 3 Short- and Long-Term Morbimortality and Development of 4 Chronic Kidney Disease in Critical Care Patients with Sepsis: 5 Prospective Cohort.’

It is a prospective cohort study of 164 patient admitted to the critical care setting and the risk of AKI outcomes is assessed through the generation of an equation that utilizes blood levels of FGF23, Erythropoeitin and Klotho.  The formula is able to predict short and long-term outcomes.

My comments are listed below:

1.       The hypothesis to apply these biomarkers in the aki setting is unclear to me.  If it is for early diagnosis, what is the lead time benefit using these biomarkers compared to AKIN criteria?  If it is for prognosis or intervention, again is there advance knowledge compared to applying serum creatinine and eGFR values?

2.       The study population selection is unclear since appears as a case control study with even division of patients.  Were severely ill patients excluded or proxy consents were utilized.

3.       How was the diagnosis of sepsis made? Blood cultures,  hemodynamics or use of vasopressors?  Were surgical or non- surgical sepsis cases included?

4.       Many of the AKI known risk factors are not stated in the demographics table.  Please explain why.

5.       The cause of AKI is not stated in the cases? Is the data available?

6.       Serum Phosphate levels are the result of Parathyroid hormone and FGF 23 changes in availability and function.  Were serum phosphate obtained?  It may be a less expensive method to obtain the same information.

7.       As a clinician, is this formula user friendly?  If information such as renal recovery from hemodialysis was obtainable, then yes.  Was time of renal recovery measured?

8.       Were the hemoglobin levels measured and would transfusions affect the erythropoietin levels?

9.       Were calcium levels measured and treated with vitamin D supplements?  Do vitamin D supplements affect FGF 23 levels?

10.   Were magnesium levels measured and treated?  Does magnesium supplementation affect Klotho levels?

Reviewer 5 Report

Dear Editor, 

I read with interest the article by Toro et al. regarding the predictive role of a combined biomarker that includes plasma fibroblast growth factor 23, erythropoietin, and Klotho in short and long-term AKI, mortality, and CKD progression in sepsis patients.

The manuscript is well written and of interest.

The introduction covers very well the current state of knowledge and supports the study.

The Materials and Methods section is well structured and easy to read.

The results and statistical analysis very well presented.

It would be very interesting if the authors had data on commonly used markers such as CPR. IL-6, including the inflammatory markers derived from the total number of neutrophils, monocytes, thrombocytes, and lymphocytes to be able to compare the marker proposed by them with those already well known and studied. I think it would offer an advantage, given that the marker proposed by the authors has a very good prediction, superior to what is found in the literature for the other markers.

Round 2

Reviewer 2 Report

Authors have performed the corrections and answered the comments in this version. Now I think this paper improved very much.

Author Response

Thanks for your favorable answer and your support.
We also agree that your contribution allows us to improve our manuscript's quality.

Reviewer 3 Report

Please note , the revised manuscript did not cover all the topics brought up by the review.  Please see below.

I reviewed the manuscript by Toro et al titled ‘A combined Biomarker That Includes Plasma Fibroblast 2 Growth Factor 23, Erythropoietin, and Klotho Predicts 3 Short- and Long-Term Morbimortality and Development of 4 Chronic Kidney Disease in Critical Care Patients with Sepsis: 5 Prospective Cohort.’

It is a prospective cohort study of 164 patient admitted to the critical care setting and the risk of AKI outcomes is assessed through the generation of an equation that utilizes blood levels of FGF23, Erythropoeitin and Klotho.  The formula is able to predict short and long-term outcomes.

My comments are listed below:

1.The hypothesis to apply these biomarkers in the aki setting is unclear to me. If it is for early diagnosis, what is the lead time benefit using these biomarkers compared to AKIN criteria?  If it is for prognosis or intervention, again is there advance knowledge compared to applying serum creatinine and eGFR values?

2.The study population selection is unclear since appears as a case control study with even division of patients.  Were severely ill patients excluded or proxy consents were utilized.

3.How was the diagnosis of sepsis made? Blood cultures,  hemodynamics or use of vasopressors?  Were surgical or non- surgical sepsis cases included?

4.Many of the AKI known risk factors are not stated in the demographics table. Please explain why.

5.The cause of AKI is not stated in the cases? Is the data available?

6.Serum Phosphate levels are the result of Parathyroid hormone and FGF 23 changes in availability and function.  Were serum phosphate obtained?  It may be a less expensive method to obtain the same information.

7.As a clinician, is this formula user friendly?  If information such as renal recovery from hemodialysis was obtainable, then yes.  Was timeof renal recovery measured?

8.Were the hemoglobin levels measured and would transfusions affect the erythropoietin levels?

9.Were calcium levels measured and treated with vitamin D supplements?  Do vitamin D supplements affect FGF 23 levels?

10.Were magnesium levels measured and treated?  Does magnesium supplementation affect Klotho levels?

Reviewer 4 Report

The authors does not answer  Q2 in report 1. No transformation was performed on the fek score.

The answer of Q1 is not that convincing. If so, the detailed information should be added in the section of study design.

Fine.
